

# Boosting aerosol surface effects: Strongly Enhanced Cooperative Surface Propensity of Atmospherically Relevant Organic Molecular Ions in Aqueous Solution

Harmanjot Kaur,[1] Stephan Thürmer,[2] Shirin Gholami,[1] Bruno Credidio,[1] Florian Trinter,[1,3] Debora Vasconcelos,[4] Ricardo Marinho,[5,6] Joel Pinheiro,[7] Hendrik Bluhm,[1] Arnaldo Naves de Brito,[7] Gunnar Öhrwall,[8] Bernd Winter,[1] and Olle Björneholm[4]*

[1]*Fritz-Haber-Institut der Max-Planck-Gesellschaft, Faradayweg 4-6, 14195 Berlin, Germany*

[2]*Department of Chemistry, Graduate School of Science, Kyoto University, Kitashirakawa-Oiwakecho, Sakyo-Ku, 606-8502 Kyoto, Japan*

[3]*Institut für Kernphysik, Goethe-Universität Frankfurt, Max-von-Laue-Straße 1, 60438 Frankfurt am Main, Germany*

[4]*Chemical and Biomolecular Physics, Department of Physics and Astronomy, Uppsala University, 75120 Uppsala, Sweden*

[5]*Institute of Physics, Brasília University (UnB), 70.919-970, Brasília, Brazil*

[6]*Institute of Physics, Federal University of Bahia, 40.170-115, Salvador, BA, Brazil*

[7]*Department of Applied Physics, Gleb Wataghin Institute of Physics, Campinas University, CEP, 13083859, Campinas SP, Brazil*

[8]*MAX IV Laboratory, Lund University, SE 22100 Lund, Sweden*

**ORCID**
*ST: 0000-0002-8146-4573*
*SG: 0000-0002-0647-1490*
*BC: 0000-0003-0348-0778*
*FT: 0000-0002-0891-9180*
*RM: 0000-0001-5854-5589*
*HB: 0000-0001-9381-3155*
*ANdB: 0000-0002-9098-444X*
*GÖ: 0000-0002-5795-8047*
*BW: 0000-0002-5597-8888*
*OB: 0000-0002-7307-5404*



*Corresponding author: Olle Björneholm: olle.bjorneholm@physics.uu.se

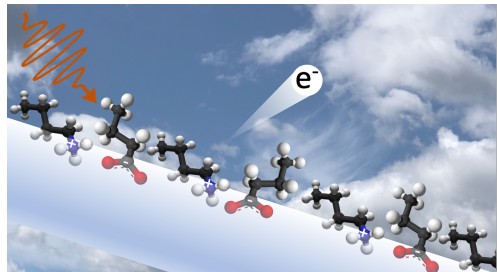

## Abstract

The effects of atmospheric aerosols are key uncertainties in climate models. One reason is the complex aerosol composition which includes a relatively large fraction of organics. Another reason is the small size of aerosols, which makes surface effects and processes important. These two factors make surface-active organics important for atmospheric aerosols, as they can affect important processes, such as chemical aging and water accommodation, as well as properties such as the surface tension, which drives droplet formation. Two important types of atmospherically relevant organics are carboxylic acids and alkyl amines, and often both are found together within aerosols. In the most atmospherically relevant pH range, these exist as alkyl carboxylate ions and alkyl ammonium ions, Using liquid-jet photoelectron spectroscopy, tuned to high surface sensitivity, we measured the alkyl carboxylate cations and the alkyl ammonium anions of alkyl chain lengths 1 to 6 carbon atoms, both as single-component and mixed-component aqueous solutions. This enabled us to systematically study how their surface propensity is affected by the length of the alkyl chains, and how cooperative ion–ion interactions result in strongly increased surface propensity. An exponential increase in surface propensity is found for the single-species solutions, with cooperative solute-solute effects in mixed solutions drastically increasing the number of molecules present at the solutions' surfaces up to a factor of several hundred. This cooperative surface propensity is shown to strongly affect the amounts of organics at the surface, with pronounced chain length-dependent variations. Our results demonstrate that the surface composition of these water-organics systems can be very different from the bulk composition, and that the surface compositions of organic mixtures cannot be directly inferred from the behaviour of the single components. For aerosols containing these or similar species, this means that all surface-related properties and processes will be enhanced, and the implication for atmospherically relevant processes such as water accommodation, droplet formation, and chemical aging, is discussed.



# Introduction


Water's liquid-vapor interface is of crucial environmental significance, considering the abundance
of liquid water covering Earth's surface, and of aqueous particles in Earth's atmosphere. The latter,
varying from microscopic aerosols to raindrops, impacts the global radiation balance by scattering
sunlight (the direct effect),(Mccormick and Ludwig, 1967) and by serving as important cloud
condensation nuclei (CCN) and ice-nucleating particles (the indirect effects).(Twomey, 1974) The
effects of aerosols have been identified by the United Nations Intergovernmental Panel on Climate
Change (IPCC) as a key uncertainty in climate models,(The Intergovernmental Panel on Climate
Change (Ipcc), 2022) and thus a better understanding of these effects is important for improving climate
modeling. Atmospheric aerosols comprise many species, including atomic ions, organic compounds
from various sources like emissions and decomposition, soot from combustion, and mineral particles.
The organic fraction, ranging from 20-90% of submicron aerosol mass, mainly forms secondary organic
aerosols (SOA) with complex compositions.(Jimenez et al., 2009) The complex mix of organic
compounds within atmospheric aerosols makes it challenging to quantify the effect of aerosols on the
climate and associated climate changes.(Kanakidou et al., 2005; Tsigaridis et al., 2014)
The importance of aerosols stems from their large surface-to-volume ratio. One notable consequence
is that the surface concentration of atmospherically relevant amino acids is up to ten times higher than
inside the aerosols.(Mocellin et al., 2017) Furthermore, the surface concentration of amino acids tends
to increase upon addition of salt to the aqueous solution.(Gopakumar et al., 2022; Björneholm et al.,
2022) Yet, existing climate models often pay less attention to aerosol surface effects because
understanding of atmospheric surface phenomena on the molecular level is lacking.(Noziere, 2016;
Lowe et al., 2019) Specifically, aforementioned organics affect surface tension, altering condensation
and evaporation rates, which in turn alters water accommodation, *i.e.*, the aggregation of water mass
onto the aerosol.(Sareen et al., 2013; Ovadnevaite et al., 2017; Davies et al., 2013; Ergin and Takahama,
2016; Miles et al., 2016; Ruehl et al., 2016) As surface species are more accessible for reactions with
atmospheric radicals, the surface propensity of constituent species also affects the aerosol chemical
aging, *i.e.*, the time evolution of the chemical composition via chemical and photochemical
processes.(McFiggans et al., 2006; Shiraiwa et al., 2011) Microscopic surface effects are thus crucial
for aerosol growth and cloud condensation nuclei activity, thereby affecting the macroscopic radiative
forcing, *i.e.*, Earth's energy balance in terms of reflection and absorption of solar radiation.
A promising experimental approach towards a deeper understanding of the molecular-scale
interfacial structure and associated processes is the application of liquid-jet photoelectron spectroscopy
(LJ-PES) to selected molecular model systems in aqueous solutions in combination with X-rays tuned
to a surface-sensitive energy range. Many important organics are amphiphilic, *i.e.*, containing both
hydrophobic and hydrophilic parts; the former often consist of non-polar groups such as alkyl chains,
while the latter consist of polar or charged groups. The surface propensity of such amphiphiles can be



seen as a balance between the hydrophilic and hydrophobic interactions, where the hydrophilic part is
solvated while the hydrophobic part tends to be only partially solvated. For example, LJ-PES studies in
conjunction with molecular dynamics (MD) simulations have previously revealed how the surface
propensity increases and how the molecular surface orientation evolves with the length of the
hydrophobic alkyl chain for alcohols and carboxylic acids.(Werner et al., 2018; Ekholm et al., 2018;
Walz et al., 2016; Walz et al., 2015; Öhrwall et al., 2015; Lee et al., 2016) This picture has been
quantitatively corroborated for perfluorinated pentanoic acid in aqueous solution, for which the distance
of different parts of the molecule from the solution surface was determined with Angstrom resolution
from the analysis of photoelectron angular distributions.(Dupuy et al., 2023) The hydrophobic ends may
undergo orientational changes, from mainly aligned parallel (to the molecular axis with respect to the
surface plane) at low surface coverage towards perpendicular with increasing coverages to make room
for more molecules. Such reorientation was, for example, observed for alcohols in aqueous
solution.(Walz et al., 2015; Walz et al., 2016)

Two common types of hydrophilic functional groups in atmospherically relevant organics are

carboxyl/carboxylate and amine/ammonium (depending on pH, -COOH/-COO$^-$, and -NH$_2$/NH$_3^+$,
respectively), and carboxylic acids and alkyl amines are examples of amphiphiles commonly found in
aerosols.(Goldstein and Galbally, 2007) Both carboxylic acid and alkyl amines are considerably surface
active.(Werner et al., 2018; Ottosson et al., 2011) Most notably, in both cases larger alkyl-chain lengths
result in larger surface propensity. However, surface propensity is also pH dependent, with a smaller
surface propensity of the charged species as compared to the neutral ones. However, as discussed above,
atmospheric aerosols often contain multiple organic species. Interaction between different organic
solutes can affect their respective surface propensity, both via cooperative or competing effects.
Cooperative ion-pairing effects at the surface in mixed hexanoate/hexyl ammonium solutions were
shown to amplify the surface propensity of both species.(Ekholm et al., 2018)

It can thus be expected that the surface propensity of atmospherically relevant alkyl carboxylate ions

and alkyl ammonium ions scales with both the length of their hydrophobic chains and cooperative ion–
ion interactions. In this work, we systematically explore how both effects jointly affect a molecule's
surface propensity, exemplified for alkyl-carboxylate and alkylammonium ions of variable alkyl-chain
lengths. Our results provide insight into the molecular driving forces affecting the surface composition
of mixed-organic aqueous solutions, which will aid atmospheric scientists in creating a parameterized
description of aerosol surface phenomena for improving climate models.

## Methods

### 1. Experiments

Most measurements of this study were performed using the *EASI* (*Electronic structure from Aqueous*

*Solutions and Interfaces*) liquid-jet photoelectron spectroscopy apparatus(Malerz et al., 2022) in tandem



with the P04 soft X-ray beamline(Viefhaus et al., 2013) of the PETRA III synchrotron radiation facility
(Deutsches Elektronen-Synchrotron DESY, Hamburg, Germany). Some measurements were repeated
at the PLÉIADES beamline of the SOLEIL synchrotron facility (Paris, France),PLÉIADES beamline
website, Synchrotron SOLEIL, Saint-Aubin, https://www.synchrotron-soleil.fr/en/beamlines/pleiades.
which is explained further below. The LJ-PES apparatus used at PETRA III is equipped with a state-
of-the-art, near-ambient-pressure hemispherical electron analyzer (HEA, Scienta Omicron HiPP-3),
complete μ-metal shielding, and large pumping capabilities for volatile liquids. Under operation
conditions, the average pressure in the interaction chamber was typically maintained at $\sim2 \times 10^{-4}$ mbar,
as accomplished with two turbomolecular pumps (with a total pumping speed of $\sim2600$ L s$^{-1}$ for water)
and three liquid-nitrogen cold traps (with a total pumping speed of $\sim35000$ L s$^{-1}$ for water). A custom-
made differential pumping chamber, installed between the interaction chamber and the last beamline
element, ensures a sufficient pressure drop across three differential stages for connection to the
beamline.(Malerz et al., 2022) The circularly polarized light from the APPLE II undulator(Sasaki,
1994) of the P04 beamline was monochromatized by a variable-line-spacing monochromator using a
1200 l/mm planar grating (9 nm groove depth, non-blazed, Au coating) and a 150 μm vertical exit-slit
opening (perpendicular to the LJ axis and the light propagation direction), adjusted by the exit slit unit
(EXSU). Photon energies of 400 eV (resolution of 70 meV) and 510 eV (resolution of 100 meV) were
employed to measure C 1s and N 1s photoelectron spectra, respectively. For a few solutes with very
high surface propensity, the C 1s PE signal could become so high to potentially saturate the detector.
This was prevented by reducing the photon flux by narrowing of the vertically oriented beam-defining
aperture (BDA), which is located 27.9 m downstream of the undulator and 43.1 m upstream of the
EXSU.(Bagschik et al., 2020) An overview of the used BDA settings, along with corresponding photon
flux values measured using a SXUV photodiode, can be found in the Supporting Information. The
beamline's vertical spot size (relevant for the LJ target) at the 150-μm EXSU opening was ~50 μm,
independent of the BDA gap, which is somewhat larger than the LJ diameter (see below). The horizontal
(along the liquid jet axis) spot size was ~180 μm. The photoelectron detection axis was at an angle of
~130° with respect to the light propagation axis, in the vertical plane, *i.e.*, the spectrometer is above and
tilted towards the beamline in the backward direction.(Malerz et al., 2022) The LJ axis is in the
horizontal (floor) plane and thus orthogonal to both the light propagation and electron detection axes.

Surface-sensitive PES measurements were performed with a photon energy resulting in a C 1s

photoelectron kinetic energy (KE) of ~100 eV, where the effective attenuation length (EAL), *i.e.*, the
probing depth into solution, is ~15 Å.(Thürmer et al., 2013) The samples were aqueous solutions of
alkyl carboxylate anions (sodium counter cation) and alkyl ammonium cations (bromine counter anion)
with variable chain lengths, with a total of 8 individual molecular species and 16 paired mixtures (see
Table 1). To keep descriptions concise, we adopt an abbreviated naming scheme, where letters A and
C refer to alkyl amines and alkyl carboxylates, respectively. In addition, we use numbers which
represent the number of carbon atoms, indicating the molecular chain length. For the former, the study



covers the methylammonium ($H_3C\text{-}NH_3^+$)$_{aq}$ cation, which is thus labeled 'A1', A2 = ethylammonium
($H_3C\text{-}CH_2\text{-}NH_3^+$)$_{aq}$, A4 = butylammonium ($H_3C\text{-}CH_2\text{-}CH_2\text{-}CH_2\text{-}NH_3^+$)$_{aq}$, A6 = hexylammonium ($H_3C\text{-}$
$CH_2\text{-}CH_2\text{-}CH_2\text{-}CH_2\text{-}CH_2\text{-}NH_3^+$)$_{aq}$, all with a $Br^-$ anion. The latter covers C1 = formate ($HCOO^-$)$_{aq}$, C2
= acetate ($H_3C\text{-}COO^-$)$_{aq}$, C4 = butyrate ($H_3C\text{-}CH_2\text{-}CH_2\text{-}COO^-$)$_{aq}$, and C6 = hexanoate  ($H_3C\text{-}CH_2\text{-}CH_2\text{-}$
$CH_2\text{-}CH_2\text{-}COO^-$)$_{aq}$, all with a $Na^+$ cation. Mixed solutions of equimolar concentration are denoted Cx /
Ay, with x, y being the carbon count. An overview of all studied single-component solutions as well as
paired mixtures with their abbreviated labels is given in Table 1; sketches of all molecules are presented
in Figure 2C. The same table format is maintained throughout the manuscript when discussing
quantitative peak intensities. Single-species solutions were prepared by dissolving methylammonium
bromide (A1), of  98% purity, ethylammonium bromide (A2), of ≥98% purity, n-butylammonium
bromide (A4), of  ≥98% purity, n-hexylammonium bromide (A6), sodium formate (C1), of  ≥99%
purity, sodium acetate (C2), of ≥99% purity, sodium butyrate (C4), of 98% purity, or sodium hexanoate
(C6), of 99-100% purity, each from Sigma-Aldrich, in demineralized water (conductivity ~0.2 µS/cm)
to yield a 0.1 M concentration. At this concentration, the surface coverage of A6 is ~0.37 and of C6
~0.15 of the maximum coverage, *i.e.*, well below surface saturation.(Ekholm, 2018) Since A6 and C6
have the highest surface propensities among the single species, all other species will exhibit a lower
surface coverage. Mixture solutions were prepared by mixing equal volumes of the pure solutions,
yielding solutions with the same total solute concentration, 0.1 M, and 0.05 M concentration for each
species.
**Table 1:** Labeling scheme for the studied molecular species with varying chain lengths: carboxylic
acids (C1-C6, vertical) and alkyl amines (A1-A6, horizontal). The eight single-component solutions
(Cx and Ay, respectively) had a solute concentration of 0.1 M. The 16 mixed solutions (Cx / Ay, italic
text) were prepared with 0.05 M of each constituent, yielding the same total concentration of 0.1 M.

<b>alkyl amine</b>

| | name | ------ | methylammonium | ethylammonium | butylammonium | hexylammonium |
|---|---|---|---|---|---|---|
| | | | label | **A1** | **A2** | **A4** | **A6** |
| **carboxylic acid** | formate | **C1** | *C1 / A1* | *C1 / A2* | *C1 / A4* | *C1 / A6* |
| | acetate | **C2** | *C2 / A1* | *C2 / A2* | *C2 / A4* | *C2 / A6* |
| | butyrate | **C4** | *C4 / A1* | *C4 / A2* | *C4 / A4* | *C4 / A6* |
| | hexanoate | **C6** | *C6 / A1* | *C6 / A2* | *C6 / A4* | *C6 / A6* |


The aqueous solutions were injected into the vacuum chamber as a liquid jet via a silica-glass
capillary nozzle with an inner diameter of 25 µm. The liquid was pumped via a Shimadzu LC-20 AD



high-performance liquid chromatography (HPLC) pump combined with an inline-degasser unit
(Shimadzu DGU-20A$_{5R}$), and then pushed through the glass capillary to yield a typical flow rate of
~0.8 ml/min. The solution temperature was kept at ~10 °C by water-cooling the LJ rod using a chiller
unit; the temperature is expected to be a few degrees lower at the point of ionization (approximately 5
mm further downstream after the liquid is expelled into vacuum) of the liquid jet, due to evaporative
cooling. At larger distances from the injection point, the liquid jet breaks up into droplets due to
Rayleigh instabilities.(Winter and Faubel, 2006) The resulting liquid spray is collected (frozen out) at
the surface of a liquid-nitrogen cold trap, downstream of the flow direction. The distance between locus
of jet – light-beam interaction and the HEA was ~0.8 mm, and the entrance aperture of the latter also
0.8 mm diameter. Accurate positioning of the jet was achieved by a high-precision x-y-z manipulator
to which the LJ assembly is mounted. The optimal overlap of all axes was continuously monitored and
adjusted during the measurement to account for signal fluctuations from small jet position drifts.
Because of stability issues noticed during analysis of the C1/A1 mixture in the first campaign,
measurements of C1, A1, and their mixture (C1/A1) were repeated at PLÉIADES. Again, photon
energies of 400 eV and 510 eV were employed to measure C 1s and N 1s photoelectron spectra,
respectively. Here, the electron spectrometer was mounted such that the electron detection axis was
perpendicular to the plane of the electron orbit in the storage ring. The LJ axis lies in the horizontal
plane (plane of the electron orbit in the storage ring). The direction of propagation of the light, the
electron detection axis and the liquid jet were mutually orthogonal to each other. The angle between the
light-polarization vector of the linearly polarized light and the spectrometer axis was set to 55º which
is close to the magic angle. A Shimadzu LC-40 AD high-performance liquid chromatography (HPLC)
pump was used to pump the liquid, and push it through a glass capillary with 40 µm orifice diameter at
a flow rate of 2.7 ml/min. The LJ is then collected in liquid form by a heated copper-beryllium catcher.
Differential pumping is achieved by housing the complete LJ assembly in an enclosure within the
vacuum chamber while using small orifices for the insertion of the X-rays and the extraction of the
photoelectrons. The distance between the LJ and the entrance of the 300 µm stainless-steel skimmer of
the spectrometer was 1 mm. For more experimental details of the setup at PLÉIADES see Refs. (Malerz
et al., 2021; Powis et al., 2015).

## 2. Data Analysis

The analysis of measured electron counts as a function of electron KE was carried out using Igor Pro
(WaveMetrics, Sutter Instrument). First, the C 1s bands were fitted for all single-species solutions, *i.e.*,
the carboxylic acids (C1 to C6) and alkyl amines (A1 to A6) to extract peak shapes and widths.
Exemplary fits for the C2 and A2 aqueous solutions are presented in Figures 1, top and center panels,
respectively. The broad, featureless signal background, originating from inelastically scattered
photoelectrons, was approximated with a linear function, which is a simplification but the most stable
choice with only two fit parameters to vary. The signal contributions from the two carbon atoms, labeled



p1 and p2 in the figure, respectively, can be separated for both solutions; all Cx and Ay species feature
two distinct carbon signal contributions as we will detail later. A noticeable asymmetry of the C 1s
bands of the carboxylic acids arises from unresolved vibrational excitation. To keep the number of
fitting parameters preferably low, we chose asymmetric exponentially modified Gaussian
(EMG)(Grushka, 1972) functions to account for the vibration contributions. The alkyl amine C 1s bands
did not show any resolvable peak asymmetry and were best fitted with two (one in case of A1) Voigt
functions instead. A Voigt function yielded a better fit than a simple Gaussian function. The added
complexity of both the EMG and Voigt functions has no impact on the results for the mixed solutions:
the shape of the EMG (asymmetry parameter $\tau$) and the Voigt function (Gaussian-to-Lorentzian width
ratio) were held fixed in subsequent fits to the spectra of the mixed solutions, which removed any
influence from these parameters. Indeed, for most of the solutions studied here, our procedure resulted
in good overall fits of the measured photoelectron spectra. There are few exceptions, where small
additional signals occur, which we attribute to contaminations of unknown origin. These features were
fitted with additional Gaussian functions (see the Supporting Information for details), but were not
included in the determination of intensities (measured as peak area) of the respective C 1s bands.
The mixed-solution spectra are fitted with a sum of the same number and type of functions as the
individual species, where the shape (asymmetry $\tau$ for the EMG and width ratio for the Voigt functions,
respectively) and peak width were kept fixed. Additional features from possible contaminants could not
be discerned (see below), and thus additional (Gaussian) functions were not included in the mixed-
solution fits. Since the contaminant features are small and not expected to be associated with a surface-
active species, their contributions to the PE spectra, if present at all, becomes diminishingly small as
peak intensities scale up rapidly for larger species. Figure 1(bottom) shows the C 1s fits for the C2/A2
mixed aqueous solution; the respective fits for the single-species solutions have been already introduced
in Figures 1 (top and center panels). The p2 bands are at similar positions for both the Cx and Ay species
and are thus strongly overlapping. In cases where both the Cx and Ay chains were present, *i.e.*, x, y > 1
for both species, the p1-p2 peak distance for Ay was set and held fixed to the result from the fit to the
single-component Ay spectra. We note that the separation into distinct peak contributions becomes
more difficult for species with increasing chain lengths. The spectral features of the chain carbons are
almost completely overlapping for all species, which is an inevitable fact of the (lack of) chemical shift.
In fact, for mixtures with the longest-chain alkyl amine A6, we had to additionally constrain the peak-
height ratio p1/p2 for the Cx component to reach a stable fit, because the signal contributions from the
chain for each constituent could not be discerned. This is a reasonable simplification since peak p1 of
the Cx species is well separated in the spectrum and can serve as an anchor for the fit to determine the
height of peak p2 for a fixed p1/p2 ratio.
We note that some of our PE spectra were unintentionally recorded under conditions of detector
saturation (see the Experiments section) which disproportionally affects the signal intensity of the
strongest bands for these spectra. Measurements of some samples (with the highest intensity) were



repeated using a lower photon flux to circumvent saturation; the procedure is described in detail in the
SI. Another complication was discovered when analyzing the peak intensities for the sodium
formate/methylammonium bromide (C1/A1) solution: We found fluctuating PE signal intensities of up
to a factor of two during the initial measurement campaign. In that case, we have repeated the
measurements from the (nominally) same C1 and A1 as well as C1/A1 concentrations in a different
measurement campaign, using a different setup at the SOLEIL synchrotron radiation facility. Those
measurements used a different sample batch and showed no sign of contamination, which reassured us
that the sidebands in the initial data originated from contaminants. The signal intensities from the repeat
measurement were scaled by the C1 signal to match the initial data, and were used instead for the results
presented here.
C 1s (relative) peak intensities, $I(Cx)$ and $I(Ay)$, the main observables in this study, which reveal a
given species' variable and competing surface propensity, were quantified by normalization to the
smallest peak-intensity value, $I(C1)$, which is from aqueous-phase formate (C1). This normalization
factor is used throughout the work for the analysis of all peak-intensity values, and thus the results
represent a relative increase in surface propensity compared to formate. Peak intensities scale with the
number of ionization targets, the photoionization cross section, and the probing depth of the C 1s
photoelectrons.(Hüfner et al., 2005) Molecular photoionization cross-sections are unknown, but are
taken to be the same for all carbon atoms. In most cases, it is useful to present the data with the
dependence on the carbon number removed, by normalizing to the number of relevant carbon sites. For
example, when discussing the total peak intensity for C4, containing four carbons, the total intensity
value is divided by four, and in the case of the C2/A4 mixture, with a total of six carbon atoms, the total
intensity will be divided by six; such normalization will be stated in the caption.
In the case of the nitrogen-containing Ay species, we also recorded and analyzed the N 1s spectra.
This procedure is much simpler, as only a single peak is present for all species, and was fitted with a
single EMG function and a linear background. No contaminants were observed here, indicating that the
contaminants are not degraded alkyl amine molecules. N 1s peak-intensity values were arbitrarily
normalized to yield the same normalized intensity value as for the C 1s of methylammonium bromide
(A1) for better comparability. Furthermore, the peak-intensity values of the mixed species must be
adjusted for differences in molecular number density since each species in the mixed solutions had a
concentration of 0.05 M instead of 0.1 M for the single-species solutions. Thus, intensity values were
adjusted by a factor of two whenever relevant for a direct comparison.
We also analyzed the valence-band PE signal intensity based on a simple height comparison of the
water $1b_1$ (HOMO) band for each solution's spectrum with that of a representative (average) neat water
spectrum; see Figure SI-2 in the SI for details.



## Results and Discussion

Figure 1 shows C 1s PE spectra for three samples, C2 (top panel), A2 (middle panel), and C2 / A2 (bottom panel). These spectra are representative of the spectra recorded for all samples listed in Table 1; all PE spectra considered in the present study including the peak fits can be found in the SI as Figures SI-4, SI-5 and SI-6. For both species, the spectra consist of two peaks: the peak p2 at the highest KE, *i.e.*, lowest binding energy (BE), corresponds to the methyl carbon. The peak p1 at lower KE, *i.e.*, higher BE, is due to ionization of the carboxylate carbon for C2, and ionization of the carbon atom next to the ammonium group in the case of A2. The chemical shifts agree well with previous studies(Ekholm et al., 2018; Ottosson et al., 2011) and can be qualitatively understood as follows. The higher BE (lower KE) of the carbon next to N and O is due to the electronegative atoms N and O reducing the electron density around the C atoms relative to the methyl carbon. The slight shift of the methyl-carbon peak between C2 and A2 is due to the different charges of the C2 and A2 molecular ions. The spectrum of the mixed C2/A2 solution can be understood as a sum of the C2 and A2 spectra, see the bottom panel of Figure 1.

We briefly comment on the definition of 'chain' length for the Cx *versus* Ay species. The fact that carbon is not part of the functional group for the Ay offsets, in practice as we see below, the chain length of this species by one with respect to Cx. Thus, we can say that A2 has a chain length of two, whereas C2 has only a chain length of one, as the carbon atom in the functional group is omitted. For this reason, we introduce the *effective* chain length $k = x-1 = y$ for the Cx and Ay species, respectively. Yet, for Ay we can still distinguish between the carbon closest to nitrogen. The intensity of peak p1 will be treated separately as needed, since it allows us to discuss molecular orientation.

### Single-component solutions

We start with the various single-component species in aqueous solution. Obtained peak intensities, based on the analysis of the C 1s and N 1s PE spectra (see Methods), are summarized in Table 2 and plotted in Figure 2A. Normalized total C 1s peak intensities, $I_{norm}$, for the carboxylate and alkyl ammonium species are plotted on a logarithmic scale against the effective chain length k, which is a measure of chain length ranging from 0 (no chain) to 6 (a six-carbon chain). Open circles represent $I_{norm}(Cx)$ and $I_{norm}(Ay)$, respectively, crosses are $I_{norm}$ of peak p1 only (related to the functional group), and triangles represent $I_{norm}$ for N 1s (Ay only). All values are normalized to the value of formate, $I(C1)$, and, as mentioned, the results can be understood as an increase in surface propensity relative to formate. The values shown in the figure are further normalized to x, y, and thus any increase is solely due to an increased surface propensity; see the bold numbers in Table 2 which are the ones plotted in Figure 2.

Formate is known to be repelled from the liquid–vapor interface,(Minofar et al., 2007) and thus can serve as a baseline for quantifying surface activity for the series of molecules studied here. We can attempt to isolate the surface contribution from the total intensity $I_{norm}$ for a quantitative characterization



of the surface composition. Since all intensities have been normalized to the number of carbon atoms
(x,y), the normalized bulk contribution should be the same for all species, *i.e.*, equal to $I_{norm}(C1)$. The
surface contribution is then obtained by subtracting $I_{norm}(C1)$ from each value of the different solutions,
$I_{surf,norm} = I_{norm} - I_{norm}(C1)$, which is equivalent to $I_{surf,norm} = I_{norm} - 1$ since all values are already
normalized by $I(C1)$. This is done for both the Cx and the Ay species, and the resulting values are
plotted in Figure 2B. Clearly, the subtraction of the bulk component is just an approximation, since the
solution–vapor interface is not a sharp transition. In fact, there is an approximately 1-nm thick gradient
over which the molecular density changes; see, for example, Refs. (Werner et al., 2018) and (Minofar
et al., 2007)  for the results of various organics. The $I_{surf,norm}$ values discussed from here on thus reflect
an average concentration within such a surface layer.

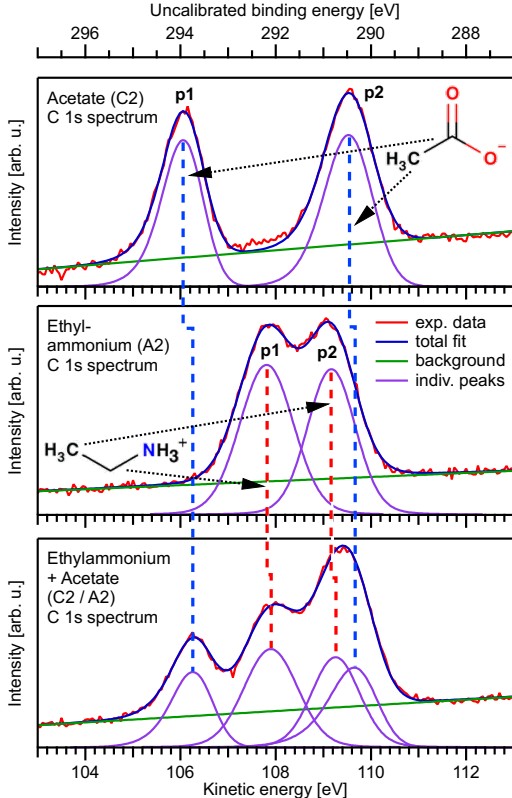


**Figure 1:** Exemplary C 1s PE spectra from aqueous solutions of 0.1 M sodium acetate (C2, top), 0.1 M
ethylammonium bromide (A2, center), and a mixture of sodium acetate and ethylammonium bromide
(C2/A2, both 0.05 M, bottom), plotted on the as-measured electron KE scale. The BE energy scale at
the top was calculated by subtracting the photon energy from the KE and was not further calibrated.
The two peaks p1 and p2 of the C2 spectrum, which are separated by a chemical shift, originate from





the methyl carbon and the carboxylate carbon, respectively. The two slightly overlapping peaks p1 and
p2 in the A2 spectrum are from the methyl carbon and the carbon closest to the ammonium group,
respectively. The spectrum of the mixed C2/A2 solution (bottom) can be understood as a sum of the C2
and A2 spectra. For details of the fitting procedure, see Methods.
Except for the smallest molecules, C1 and A1, peak areas increase approximately exponentially with
increasing k (note the logarithmic ordinate scale of Figure 2A), demonstrating a strong increase in
surface propensity as the hydrophobic chain is extended. This is qualitatively expected given the well-
known hydrophobicity of carbon chains. For x = y, the alkyl ammonium ions (Ay) have a somewhat
higher surface propensity than the alkyl carboxylate ions (Cx), in agreement with the results for C4, C5,
A4, and A6 in Ref. (Werner et al., 2018). Note again that the chain length for Ay *includes* the carbon
next to the functional group, while for Cx the carbon inside the functional group is excluded to calculate
the k values. A good match between Cx and Ay demonstrates that indeed the effective chain length and
not the absolute number of carbons determines the surface propensity of these species.
**Table 2:** Analysis results for each single carboxylic acid (Cx) and alkyl amine (Ay) solution extracted
from peak fitting to the C 1s PE spectra. All peak intensities are normalized to the value of formate
(C1); see Methods. Peak intensities increase approximately exponentially with increasing chain length
for both the Cx and Ay species; compare Figure 2. The peak-intensity ratio R between peaks p1 and p2
(see Figure 1) is compared to $R_{ideal}$, calculated from the number of carbons contributing to peak p2; only
one carbon atom contributes to peak p1 for all species. The results reveal a deviation from unity for all
species. $R / R_{ideal} > 1$ ($R / R_{ideal} < 1$) indicates a preferable orientation with the chain (the functional
group) towards the surface. The bottom-most row reports the relative change in the valence-band signal
of each solution compared to neat water; see also Figure SI-2. Bold values are plotted in Figure 2, panels
A and B, respectively.

| sample | C1 | C2 | C4 | C6 | A1 | A2 | A4 | A6 |
|---|---|---|---|---|---|---|---|---|
| **effective chain length k** | **0** | **1** | **3** | **5** | **1** | **2** | **4** | **6** |
| **intensity $I_1$ of peak p1** | **1.00** | **1.08** | **2.49** | **6.75** | **1.20** | **1.85** | **6.76** | **20.55** |
| intensity $I_2$ of peak p2 | --- | 1.31 | 9.01 | 45.9 | --- | 1.61 | 18.7 | 104.6 |
| total: $I_{tot} = I_1 + I_2$ | 1.00 | 2.38 | 11.5 | 52.7 | 1.20 | 3.46 | 25.5 | 125.1 |
| **carbon-normalized $I_{norm} = I_{tot}/(x,y)$** | **1.00** | **1.19** | **2.87** | **9.25** | **1.20** | **1.73** | **6.37** | **20.86** |
| total at surface: $I_{surf} = I_{tot} - I_{tot}(C1)$ | 0 | 1.38 | 10.5 | 51.7 | 0.20 | 2.46 | 24.5 | 124.1 |
| **normalize: $I_{surf,norm} = I_{norm} - I_{norm}(C1)$** | **0** | **0.19** | **1.87** | **8.25** | **0.20** | **0.73** | **5.37** | **19.86** |
| peak-intensity ratio $R = I_2 / I_1$ | --- | 1.21 | 3.63 | 6.80 | --- | 0.87 | 2.77 | 5.09 |
| ideal ratio $R_{ideal}$ (x-1 / y-1) | --- | 1 | 3 | 5 | --- | 1 | 3 | 5 |
| *ratio deviation R / $R_{ideal}$* | --- | *1.21* | *1.21* | *1.36* | --- | *0.87* | *0.92* | *1.02* |





| | relative valence-band signal | 0.96 | 0.97 | 0.94 | 0.85 | 0.99 | 0.92 | 0.91 | 0.87 |
|---|---|---|---|---|---|---|---|---|---|


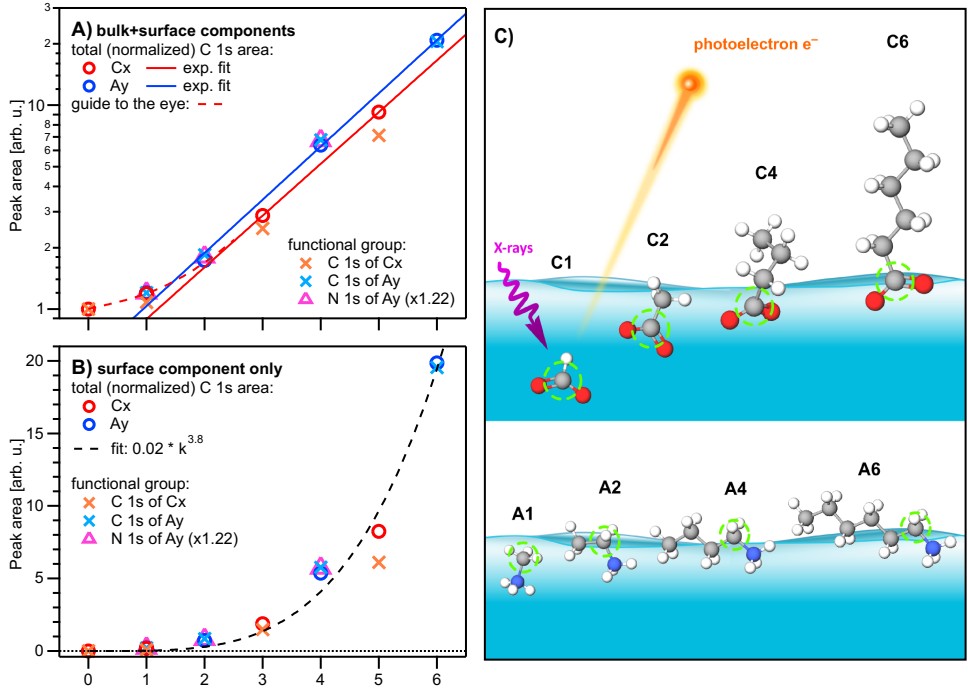


**Figure 2: (A)** Peak areas extracted from the fits to the PE spectra for each single-component solution of carboxylic acid and alkyl amine (Cx and Ay, respectively; x,y = 1,2,4,6), normalized by the value of formate, I(C1), and the number of carbons sites (x, y) within each molecule. Panel **(B)** shows the data of Panel (A) after additional subtraction of I(C1) which represent the surface contributions (see text); note the different vertical axis scales. All values are plotted against the effective chain length k (bottom axis). Red and blue open circles represent the total-area values $I_{tot} = I_1 + I_2$, *i.e.*, a sum of all C 1s intensities, of the Cx and Ay species, respectively. Orange and light-blue crosses represent C 1s intensities of only peak p1 (related to the functional group) for Cx and Ay, respectively (compare Fig. 1). The N 1s peak intensities of Ay are plotted as purple triangles, which coincide with the crosses for Ay (*i.e.*, the carbon near the functional group) when scaled with an arbitrary factor of 1.22. Both Cx and Ay show an approximately exponential increase as a function of k; note the logarithmic scale in panel (A) and the red and blue lines, which are exponential fits to the Cx and Ay data, respectively. Both species deviate somewhat from the exponential trend at low k, indicating a weaker promotion of surface propensity by short chains; see the red dashed curve as a guide to the eye. Small deviations of the crosses (functional group) above (Ay) or below (Cx) the circles (total intensity) values are due to molecular orientation (see text). **(C)** Sketch of the likely average depth and orientation of each species



as inferred from the absolute and relative intensities. The carbon site producing peak p1 is marked with
a green circle.

The surface contributions (Figure 2B) for both species exhibit similar behavior and can be

approximately described by $I_{surf,norm} \approx 0.02 \, k^{3.8}$. This is an arbitrary function obtained by fitting the data
without any theoretical justification. Yet, we would like to showcase the possibility of a parametrized
description of surface propensities, which would foster an inclusion of surface phenomena in improved
atmospheric models. We have also analyzed corresponding intensity changes of the solvent, *i.e.*, the
reduction of (water) valence-band PE signal intensity as a function of x,y compared to an average neat-
water valence spectrum. Results are shown in Figure SI-2B.

From the relative intensities between peaks p1 and p2, *i.e.*, the intensity originating from the carbon

close to or within the functional group $I_1$ relative to that from the chain carbons $I_2$, we can also obtain
information about the average molecular orientation at the surface. Returning to Figure 2A, we take a
closer look at the peak intensities, $I_1$ (crosses), in close relation to the functional group; this carbon site
can be easily identified in the C 1s PE spectra due to its associated large chemical shift. A similar but
not exactly matching trend to the normalized total peak intensity, $I_{norm}$, is observed. For Ay, $I_1$ values
tend to be slightly higher than $I_{norm}$ (compare circles vs. crosses); this is mirrored in the behavior of the
N 1s data (triangles). For Cx, the $I_1$ values tend to be somewhat lower than $I_{norm}$. Both effects can be
interpreted to originate from molecular orientation: if one end of the molecule is closer to the solution–
vapor interface, its signal will be larger compared to other molecular sites, which are pointing further
into the bulk solution. We quantify this by calculating the intensity ratio R between the peaks, *i.e.*, R =
$I_2 / I_1$. This ratio can then be compared to $R_{ideal}$, calculated from the number of carbons contributing
only to peak p2, *i.e.*, the chain. If R = $R_{ideal}$ then all carbon sites are exposed equally (they are at equal
probing depths) on average, implying that the molecules are either parallel to the surface along their
long axis or randomly oriented. If instead R / $R_{ideal} \neq 1$, the molecule is preferably oriented normal to
the interface (anchored) with one end. Table 2 (bottom part) summarizes the values of R, $R_{ideal}$, and R /
$R_{ideal}$ for each species. It is apparent that the ratio R / $R_{ideal}$ is consistently above unity for Cx, which
implies that the Cx molecules are oriented with the hydrophilic functional group towards the bulk
solution, and the hydrophobic chain towards the vacuum. For Ay, the opposite trend is observed: R /
$R_{ideal}$ is slightly smaller than unity. This is surprising since it implies that the (carbon near the) functional
group is closer to the interface than the chain. That is, the molecule lies rather parallel in the interfacial
plane, despite the hydrophilic interaction of the amine end. The proposed orientations and relative
depths of both species are sketched in Figure 2C.

To summarize, the single-component aqueous solutions of carboxylate anions and alkyl ammonium

cations show an approximately exponentially increased surface propensity as a function of length of the
hydrophobic alkyl chain. For the *same number* of carbon sites (*i.e.*, x = y), the surface propensity is
higher for alkyl ammonium cations than for carboxylate anions, consistent with the larger *effective* alkyl



chain length k of the former (k = x-1 = y) and previous results for C4, C5, A4, and A6 in Ref. (Werner
et al., 2018). Moreover, the carboxylate anions seem to have an orientation perpendicular to the surface
plane, whereas the alkyl ammonium cations lie parallel to the surface plane

**Mixed solutions**

We now turn to the mixed solutions, with results summarized in Table 3 and plotted in Figure 3.
Analogous to Figure 2, Figure 3 shows the total normalized intensity $I_{norm}$ in panel (A) and the surface
contribution $I_{surf,norm}$ in panel (B), with the latter also summarized in the table. To emphasize the changes
in surface composition, the normalized total surface intensity, $I_{surf,norm}$, a measure of the combined
amount of organic molecular ions at the surface, is derived as before for the single-component solutions
as $I_{surf,norm} = I_{norm} - I_{norm}(C1)$. In the figure, results for the Cx and Ay single-species solutions are
highlighted by the red and blue circles, respectively. Values vastly increase for the mixed solutions,
Cx/Ay. Comparing $I_{surf,norm}$ for the two mixtures of the smallest, C1/A1, and largest molecules in this
study, C6/A6, we find an increase by a factor of ~230. Neglecting depth-distribution differences, one
can directly relate this to the different number of molecular ions at the surface.
To quantify cooperative effects, we can make the ansatz that in absence of such effects the intensity
should just be the sum of the individual species' intensity, $I_{sum} = I(Cx) + I(Ay)$. We then compare this
with the measured intensities for each mixture by calculating the ratio $R_{coop} = I_{surf,norm} / I_{sum}$; the resulting
values are summarized in Table 4. A ratio larger than unity corresponds to a larger than expected surface
propensity. This is more and more the case towards longer chain lengths, *i.e.*, higher x,y values. The
increase in the mixed solutions clearly shows that ion–ion interactions lead to a cooperative surface
enrichment of the organic molecular ions. We can discern roughly two regimes analogous to the single-
species results: a slow increase and small cooperative effect when the chain is short (absent) and a large
effect for long chains.  For mixtures with C1 and C2 only an insignificant cooperative effect is observed;
C2/A1 is an exception, but we assume this is an outlier produced by a too high relative signal intensity
for this mixture in the experiment. For larger x,y the increase is more pronounced, reaching up to a
factor of three for C6/A4. We would expect the ratio for the C6/A6 mixture to be higher, but it is
possible that the surface already becomes saturated with molecules in this mixture, leading to a
relatively small increase as compared to the (already very surface-active) individual species C6 and A6.
We would like to emphasize at this point that surface saturation is another crucial aspect determining
the availability of molecular ions at the surface; here, saturation plays a limiting role for enrichment.
We have seen above that cooperative effects can multiply the number of molecules at the surface by a
factor of several hundred, which can quickly saturate the surface even at small bulk-solution
concentrations. Thus, the relative increase in number density may be much larger at small initial
concentrations very far from saturation, while only a small or even no enrichment may be observed for
an already relatively high initial concentration of each constituent species. Surface saturation should
thus always be considered when modeling ion densities. We also note that an asymmetric mixture



(deviations from the 1:1 concentration ratio) may further complicate the interaction, which is, however,
beyond the current study.
**Table 3:** Total surface intensity, $I_{surf,norm}$, of all C 1s peaks and species combined, *i.e.*, the sum of all
C 1s peaks not separated into different molecular sites, which were extracted from fits to PE spectra of
single- (frame) and mixed-species (italic text) solutions; the former values are included for comparison
and are the same as in Table 2. All peak intensities were normalized to the value of formate (C1) and
to the relevant number of carbons (see Methods). Furthermore, values have been adjusted for
differences in molecular number density, *i.e.*, 0.05 M (mixtures) *versus* 0.1 M (single species). The
error for all values is estimated to be ±0.05 from intensity fluctuations and fit errors.

|  |  |  | **A1** | **A2** | **A4** | **A6** |
|---|---|---|---|---|---|---|
|  | **single** | ----- | 0.20 | 0.73 | 5.37 | 19.86 |
|  | \| | **mix:** | \| | \| | \| | \| |
| **C1** | 0.00 | ---- | *0.19* | *0.88* | *5.94* | *22.92* |
| **C2** | 0.19 | ---- | *0.73* | *1.05* | *6.53* | *22.24* |
| **C4** | 1.87 | ---- | *2.65* | *3.64* | *13.58* | *34.91* |
| **C6** | 8.25 | ---- | *14.66* | *19.31* | *38.93* | *43.85* |


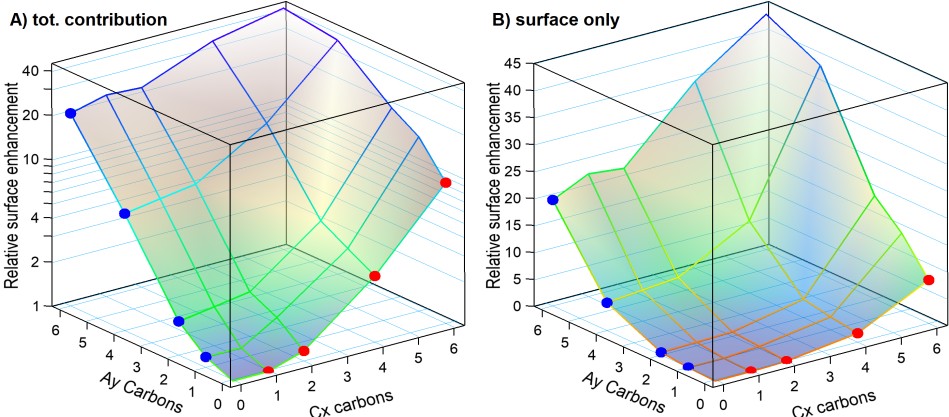


**Figure 3: (A)** Normalized total C 1s peak intensity, $I_{norm}$, *i.e.*, divided by the total number of carbon
atoms and by the value of C1, for all studied species on a log scale. **(B)** Total surface component only,
$I_{surf,norm}$, *i.e.*, after subtraction of I(C1), on a linear scale. The two abscissae represent the total number
of carbons (x,y) in the molecule for Cx and Ay, respectively. Red and blue circles mark the values for
each single-species solution Cx and Ay, respectively, and correspond to the red and blue circles in





Figures 2. The plots can be seen as analogous to Figures 2A and 2B but now including the mixed
solutions as a pseudo-3D representation. The highest overall surface propensity is observed for C6/A6.
**Table 4:** Surface enhancement ratio $R_{coop} = I_{surf,norm} / I_{sum}$ for the mixed solutions relative to the sum of
the individual species' intensity $I_{sum}(Cx/Ay) = I_{surf,norm}(Cx) + I_{surf,norm}(Ay)$, using $I_{surf,norm}$ from Table 3.

|    | A1 | A2 | A4 | A6 |
|----|----|----|----|----|
| C1 | *0.95* ± 0.29 | *1.21* ± 0.12 | *1.11* ± 0.05 | *1.15* ± 0.05 |
| C2 | *1.69* ± 0.23 | *1.14* ± 0.10 | *1.17* ± 0.05 | *1.11* ± 0.05 |
| C4 | *1.28* ± 0.07 | *1.40* ± 0.06 | *1.88* ± 0.05 | *1.61* ± 0.05 |
| C6 | *1.84* ± 0.05 | *2.27* ± 0.06 | *2.96* ± 0.06 | *1.59* ± 0.05 |


**Microscopic mechanism for cooperative surface enrichment**

When only a single molecular ion species is present, the molecular ions on the surface repel each
other via their charged headgroups while their inorganic counter ions are located beneath the surface
layer, as schematically illustrated in Figure 4. Coulomb repulsion makes a high surface coverage of
molecular ions energetically unfavorable. However, in the mixed-solute systems, Coulomb repulsion is
reduced as the alkyl-ammonium cations and alkyl-carboxylate anions act as counter ions for each other,
fostering a cooperative effect that allows for greater coverage of organic molecular ions at the surface.
Cooperative surface enrichment similar to the observation in the present study has been reported for the
C6/A6 system,(Ekholm et al., 2018) and was qualitatively attributed to a combination of ion-pairing
between the charged functional groups of the respective organic ion, hydrophobic expulsion of the alkyl
chains from the surface, and van der Waals interactions between the alkyl chains. Furthermore, the close
packing of the alkyl chains contributes to the effect: Molecules align perpendicular to the surface and
are stabilized by van der Waals interactions between the chains, analogous to alcohols.(Walz et al.,
2015; Walz et al., 2016)

**Surface orientation**

For the single-species solutions, we concluded from the C 1s peak-intensity ratios, $R = I_2/I_1$, between
intensities originating from the carbon close to or within the functional group $I_1$ relative to that from the
chain carbons $I_2$, that the Cx anions seem to have an average orientation perpendicular to the surface
plane, whereas the Ay cations are rather lying parallel to the surface plane. What then is the molecular
orientation in the mixed cases, considering the much higher molecule number densities at the surface?
In Table 5, we present the C 1s peak-intensity ratios R and $R / R_{ideal}$ as defined for the single-component
cases. Again, a value of $R / R_{ideal}$ above (below) 1 indicates a preferable orientation with the chain (the
functional group) closer towards the surface. We observe that the Cx species retain their preferential





perpendicular orientation, as expected. Similarly, the Ay species largely maintain their preferentially
parallel orientation for the most part. Interestingly, the data shows Ay changing into a perpendicular
orientation for C6/A4, C6/A6, and C4/A6, which hints at a configuration that is normal to the surface
and thus aligned with Cx at the surface. Note however, that the results for the latter two cases are less
reliable, since the peak ratio $I_2/I_1$ was constrained for the Cx component, thus possibly arbitrarily
inflating the ratio for Ay. Still, such a result may not be unexpected considering that a close packing of
aligned molecules would use the available space more effectively (compare Figure 4, bottom panel).
This result may also be related to the particular concentration likely reaching surface saturation, which
has the tendency to force molecules into an aligned configuration.

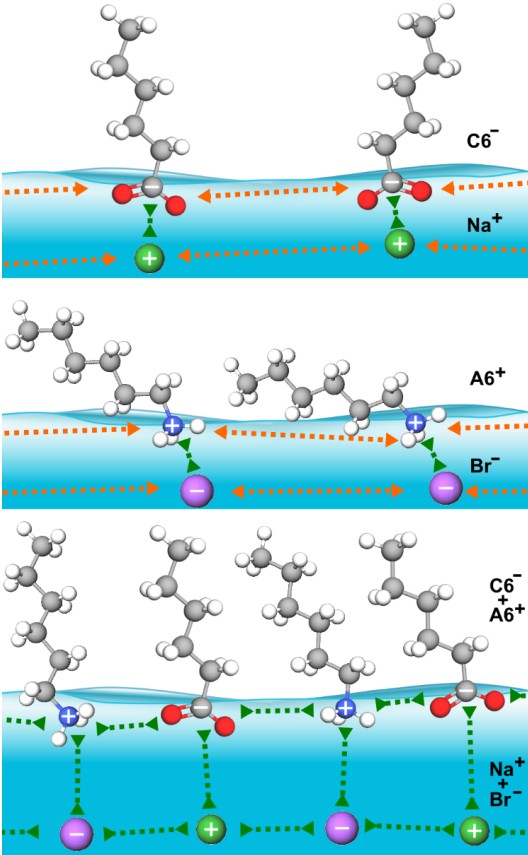


**Figure 4:** Schematic illustration of attractive (green arrows) and repulsive (orange arrows) interactions
between the organic and inorganic ions in the surface region for the C6 (top), A6 (middle), and C6/A6
(bottom) cases.



**Table 5:** Relative C 1s peak-intensity contribution from the chain $I_2$ *versus* the functional group $I_1$ for
(A) the Cx and (B) the Ay species in aqueous solution. $I_1$ and $I_2$ of each species were extracted separately
from the combined PE signal *via* fitting, and $R = I_2/I_1$ was calculated. The result is compared against
$R_{ideal}$ = x-1,y-1, calculated from the number of carbons which contribute to $I_2$. Below each entry we also
present **$R / R_{ideal}$** in bold. A value of $R / R_{ideal} > 1$ ($R / R_{ideal} < 1$) indicates a preferable orientation with
the chain (the functional group) closer towards the surface. Values marked with a star (*) in panel (A)
for mixtures of Cx with A6 are no fit results, since these values were constrained during the fitting to
reach a stable outcome for the strongly overlapping p2 peaks of Cx/A6; values were chosen to represent
averages of the results from the other Cx/Ay mixtures with y < 6.

| (A) Cx | $R_{ideal}$ | no Ay | A1 | A2 | A4 | A6 |
|---|---|---|---|---|---|---|
| C2 | 1 | R = 1.21 | 1.39 | 1.24 | 1.14 | 1.25* |
|  |  | $R/R_{ideal}$ = 1.21 | 1.39 | 1.24 | 1.14 | 1.25* |
| C4 | 3 | 3.63 | 3.67 | 3.63 | 4.27 | 3.79* |
|  |  | 1.21 | 1.22 | 1.21 | 1.42 | 1.26* |
| C6 | 5 | 6.80 | 6.81 | 6.76 | 6.78 | 6.80* |
|  |  | 1.36 | 1.36 | 1.35 | 1.36 | 1.36* |


| (B) Ay | $R_{ideal}$ | no Cx | C1 | C2 | C4 | C6 |
|---|---|---|---|---|---|---|
| A2 | 1 | R = 0.87 | 0.84 | 0.82 | 0.88 | 1.00 |
|  |  | $R/R_{ideal}$ = 0.87 | 0.84 | 0.82 | 0.88 | 1.00 |
| A4 | 3 | 2.77 | 2.76 | 2.82 | 2.76 | 3.69 |
|  |  | 0.92 | 0.92 | 0.94 | 0.92 | 1.23 |
| A6 | 5 | 5.09 | 4.49 | 4.68 | 6.14 | 9.91 |
|  |  | 1.02 | 0.88 | 0.94 | 1.23 | 1.98 |


**Surface composition**

We discussed that the surface propensity of the single species increases with the number of carbons
x,y, and that cooperative ion–ion interactions lead to an additional increase of surface propensity for
the mixed solutions. How do these two effects combined influence the relative amounts of Cx and Ay
at the surface? In Table 6, we summarize the ratio $I_{surf,norm}(Ay)/I_{surf,norm}(Cx)$, *i.e.*, the ratio of the total
intensities for each species, which is an indirect measure of the amount of Ay molecules relative to the
amount of Cx molecules at the surface. We observed that the ratio is larger than unity when the Cx
moiety has a short (C2) or no chain (C1), *i.e.*, the surfaces of these solutions are dominated by the Ay
species. For Cx with longer chains, C4 and C6, combined with short-chained Ay, A1 and A2, the
situation is reversed, *i.e.*, the ratio is smaller than unity. If the chains of both molecules are long, then



the ratio converges to 1 (a 1:1 molecule ratio), which is expected when considering that a mutual charge
neutralization would favor equal amounts of cationic Cx and anionic Ay species at the surface. Note
that A6 is a longer molecule than C6 because of the nitrogen in the functional group, which explains
the larger than unity intensity ratio for Cx/A6 (with x = 4,6), *i.e.*, the A6 molecule is likely protruding
out further when both are aligned upright at the surface. We conclude that the species with the longer
chain dominate the surface of the mixed solutions, and if both species contain long carbon chains, they
are present in approximately equal amounts.
**Table 6:** Relative surface contribution of Ay *versus* Cx to the C 1s PE spectra of the mixed solutions,
*i.e.*, an intensity ratio obtained as $I_{surf,norm}(Ay)/I_{surf,norm}(Cx)$. Errors are calculated via error propagation,
and can get large if the denominator is very small (such as for C1/A2).

|     | A1             | A2              | A4             | A6             |
|-----|----------------|-----------------|----------------|----------------|
| C1  | 2.12 ± 0.94    | 21.1 ± 17.4     | 4.55 ± 0.15    | 3.35 ± 0.05    |
| C2  | 1.27 ± 0.13    | 1.88 ± 0.15     | 2.77 ± 0.05    | 2.45 ± 0.05    |
| C4  | 0.57 ± 0.05    | 0.84 ± 0.05     | 1.09 ± 0.05    | 1.46 ± 0.05    |
| C6  | 0.68 ± 0.05    | 0.69 ± 0.05     | 1.11 ± 0.05    | 1.60 ± 0.05    |


**Amount of carbon at the surface**

So far, we discussed the number density of organic molecular ions at the surface. However, some
atmospherically relevant aspects, such as the availability of carbon for reactions with incoming radicals
and the effects on water accommodation, rather scale with the absolute amount of carbon.(Davies et al.,
2013; Ergin and Takahama, 2016; Miles et al., 2016; Ruehl et al., 2016; Shiraiwa et al., 2011) Here, the
total C 1s surface intensity, $I_{surf} = I_{tot} - I_{tot}(C1)$, *i.e.*, not normalized by the number of carbons (x,y),
provides a measure of how the amount of surface carbon varies. These values are summarized in Table 7.
Since molecular ions with higher surface propensity also tend to contain more carbon atoms, the amount
of carbon at the surface scales even stronger with the alkyl chain length than the amount of organic
molecular ions itself. For example, the relative amount of carbon at the surface is ~1400 times higher
for C6/A6 than for C1/A1.
**Table 7:** Relative amounts of carbon at the surface, expressed as the total intensity minus the bulk
contribution $I_{surf} = I_{tot} - I_{tot}(C1)$.

|          | A1      | A2   | A4   | A6   |
|----------|---------|------|------|------|
| single   | -----   | 0.20 | 2.46 | 24.5 | 124 |
| mix:     |         |      |      |      |



| | | | | | | |
|---|---|---|---|---|---|---|
| **C1** | 0 | ---- | *0.19* | *1.82* | *16.4* | *82.8* |
| **C2** | 1.38 | ---- | *1.49* | *3.10* | *21.6* | *92.0* |
| **C4** | 10.5 | ---- | *8.11* | *12.9* | *57.3* | *179* |
| **C6** | 54.5 | ---- | *53.8* | *80.3* | *199* | *268* |


**Implications for inorganic ions**

Another aspect of the surface enrichment of organic molecular ions concerns their ability to draw inorganic ions to the surface. Inorganic ions such as halides are important in atmospheric chemistry, as exemplified by the ozone depletion through a reaction with iodide and bromide,(Moreno et al., 2018; Chen et al., 2021) the production of $Cl_2$ from OH (gas) and $Cl^-$(aq),(Laskin et al., 2006) and the reaction between $N_2O_5$ (gas) and $Br^-$(aq).(Sobyra et al., 2019) These reactions involve a gas-phase species and a solvated halide ion, hinging on the presence of the latter at the surface. In single-solute solutions, the inorganic ions act as counterions to the surface-enriched organic ions, which leads to the formation of an electric double layer where organic ions occupy the surface and inorganic ions reside in a sub-layer underneath. The considerable enrichment of either positively or negatively charged organic ions on the surface has been shown to lead to a notable increase in the concentration of inorganic counterions within this sub-layer.(Gopakumar et al., 2022) However, in mixed cationic–anionic molecular ion solutions, the inorganic ions are not the main counter ions of the organic ions, as discussed above. As a result, the inorganic counter ions can be expected to exhibit reduced enrichment in the sub-layer of such mixed cationic–anionic molecular ion solutions (compare Figure 4), and hence fewer ions are available for reactions with gas-phase species.

**Atmospheric implications**

Organic matter is ubiquitous in atmospheric aerosols, both on land and in the ocean, from the tropics to the Arctic. Many organic substances are surface active to a certain extent and also contain (de)protonatable groups such as amino and carboxyl groups, forming molecular ions in a broad pH range from slightly below 5 to somewhat above 10. On the microscopic level, the surface composition of aqueous aerosols has been discussed in terms of solvent-solute interaction, and we show here that solute-solute interactions can also substantially increase the amounts of organics at the surface.

The presence of an outer organics-enriched layer has been shown to influence a number of relevant properties and processes, such as optical properties and shortwave radiative effects, water accommodation, and chemical aging, see for example, Refs. (McFiggans et al., 2006; George et al., 2010; Shiraiwa et al., 2011; Sareen et al., 2013; Davies et al., 2013; Ergin and Takahama, 2016; Miles et al., 2016; Ruehl et al., 2016, Ovadnevaite et al., 2017; Lowe et al., 2019).



Organics at the surface lower the surface tension, which is most relevant for the present study. This
directly affects aerosol droplet formation, as described by classic Köhler theory (Köhler, H. 1936;
McFiggans et al., 2006), and leads to significant enhancements of cloud condensation nuclei (Sareen et
al., 2013; Ovadnevaite et al., 2017). Furthermore, the amphiphilic organics at the surface tend to be
oriented with the carbon chains outwards. These organics can form a hydrophobic film, which, on the
microscopic scale, will reduce the sticking coefficient of incoming water molecules and thereby affect
water accommodation, as well as reducing the frequency of water molecules leaving the liquid phase,
i.e., reduce evaporation (McFiggans et al., 2006; Davies et al., 2013; Ergin and Takahama, 2016; Miles
et al. 2016; Ruehl et al., 2016). Yet another aspect is that surface species are more accessible than bulk-
solvated species for reactions with atmospheric radicals. The cooperatively enhanced surface propensity
sets the stage for further chemistry, as surface species are chemically more active than those in the
aerosol bulk. This affects e.g. the aerosol chemical aging, *i.e.*, the time evolution of the chemical
composition via chemical and photochemical processes (McFiggans et al., 2006; George et al., 2010;
Shiraiwa et al., 2011). These three examples illustrate how surface enrichment of organics influences
atmospherically fundamental surface properties and processes.
On the microscopic scale, many common amphiphilic organics containing amino and carboxyl groups
are strongly surface enriched by solute-solvent interactions, implying that modeling aqueous aerosols
as homogenous droplets would be inadequate for surface-related phenomena. Our present results show
that the surface propensity can be further strongly enhanced in a wide and environmentally relevant pH
range by solute-solute interactions, mainly between the oppositely charged molecular ions. This implies
that to properly model the surface composition of aqueous aerosols, and hence all surface related
properties and processes, such cooperative effects boosting the single-solute surface propensity would
have to be considered.
On the macroscopic scale, these changes in surface composition can, therefore, significantly influence
radiative forcing *via* aerosol growth, cloud condensation nuclei activity, and aerosol chemical aging.
Our results demonstrate the principle feasibility of a more advanced input for creating parameterized
descriptions of aerosol surface composition needed to properly account for their impacts in climate
models. Specifically, the observed drastic increase in surfactant density due to the molecular
interactions between different types of organic surfactants would be one effect to be included in future
modeling, e.g., cloud droplet formation.

## Conclusions

The ionic alkyl amines and carboxylic acids, crucial in the atmosphere as organic compounds, are
prevalent over their non-ionic forms in solutions at pHs near 7. We investigated the composition of
surfaces in aqueous solutions containing single-component as well as mixtures of the carboxylic acid



cations formate, acetate, butyrate, hexanoate and the alkyl amine anions methylammonium,
ethylammonium, butylammonium, hexylammonium, relevant in an atmospheric context. By using
surface-sensitive X-ray-based PES measurements, we show that mixtures of these compounds exhibit
a notable surface enrichment in organic ions compared to solutions with just one species. The
availability of molecular ions at the surface scales exponentially as a function of carbon chain length,
yielding an increase of up to a factor ~230 of the molecular number density, and ~1400 times the amount
of carbon between mixtures of the smallest species and the largest species studied here. This enrichment
arises mainly from ion-pairing interactions of the two ionic species, even at low bulk concentrations.
Yet, surface saturation imposes a limit on the maximum achievable enrichment. From this result, it is
anticipated that even small variations in composition with admixture of different species can lead to
significant changes of atmospherically relevant surface properties and processes such as surface tension,
condensation rates, evaporation rates, water accommodation, and the chemical aging of aerosols.
Furthermore, changes in the surface composition and condition may significantly impact radiative
forcing at a larger scale *via* aerosol growth and cloud condensation nuclei activity. Our findings
underscore the necessity for a comprehensive understanding of the surface composition of aqueous
solutions of organic molecules, which is a critical aspect for enhancing the accuracy of aerosol modeling
within climate models.

## Author Contributions

G. Ö. and O. B. conceived the experiments. H. K., S. G., B. C., F. T., D. V., R. M., J. P., H. B., A. N.
B., G. Ö., B. W., and O. B. planned, prepared, carried out the experiments, and discussed the data. H.
K. and S. T. analyzed the data. S. T., B. W., and O. B. wrote the manuscript with feedback from all
authors.

## Data Availability

The data of relevance to this study have been deposited at the following DOI:
10.5281/zenodo.12644491.

## Conflicts of interest

There are no conflicts to declare.



## Acknowledgements

We acknowledge DESY (Hamburg, Germany), a member of the Helmholtz Association HGF, for the provision of experimental facilities. Parts of this research were carried out at PETRA III, and we would like to thank Moritz Hoesch and his team for assistance in using beamline P04. Beamtime was allocated for proposal I-20220937 EC. H.K. and B.W. acknowledge funding from the European Research Council (ERC) under the European Union's Horizon 2020 research and innovation programme (grant agreement No. 883759, AQUACHIRAL). S.T. acknowledges support from the JSPS KAKENHI Grant No. JP20K15229 and ISHIZUE 2024 of Kyoto University. F.T. acknowledges funding by the Deutsche Forschungsgemeinschaft (DFG, German Research Foundation) - Project 509471550, Emmy Noether Programme. F.T. and B.W. acknowledge support by the MaxWater initiative of the Max-Planck-Gesellschaft. O.B. acknowledges support from the Swedish Research Council (VR) through Project 2023-04346 and the Swedish Foundation for International Cooperation in Research and Higher Education (STINT) through Project 202100-2932. R.M., J.P., and A.N.B. acknowledge support from the Swedish-Brazilian collaboration STINT-CAPES process no. 88881.465527/2019-01. ANB acknowledges support from FAPESP (the Sao Paulo Research Foundation, Process number 2017/11986-5) and Shell and ANP (Brazil's National Oil, Natural Gas and Biofuels Agency); and CNPq-Brazil process 401581/2016-0.

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
