# Peer review of "Boosting Aerosol Surface Effects: Strongly Enhanced Cooperative Surface Propensity of Atmospherically Relevant Organic Molecular Ions in Aqueous Solution"

_EGUsphere, 2024_

## Referee Comment (RC1)

**Comments on "Boosting aerosol surface effects: Strongly Enhanced Cooperative Surface Propensity of Atmospherically Relevant Organic Molecular Ions in Aqueous Solution" by Harmanjot Kaur et al.**

The paper investigates the surface propensity of organic surfactant molecular ions in aqueous solutions, which is crucial for understanding atmospheric aerosol behaviour. It focuses on carboxylic acids and alkyl amines, common organics surfactants in aerosols, and measures their surface enrichment using liquid-jet photoelectron spectroscopy. The study reveals an exponential increase in surface propensity with chain length and a significant boost in surface enrichment due to cooperative ion-ion interactions in mixed solutions. Results show surface compositions can vastly differ from bulk, impacting atmospheric processes like droplet formation and chemical aging. This research enhances the understanding of aerosol surface phenomena needed for climate model. The topic fits scope of ACP. The paper can be considered for publication after revisions that address the following concerns.

**Major comments:**

**(1)** As the authors described that the experiments were all performed under 10 °C (solution), $2 \times 10^{-4}$ mbar (environment), So how do the authors think that the experiments could provide valuable information for atmospheric implication. I suggest the author give more explanations on temperature setting and improved the atmospheric implication part.

**(2)** As reported in the literature (for example, Figure 1 in Noziere, Baduel, and Jaffrezo (2014), surfactant delayed surface tension equilibrium, indicating that bulk-to-surface partitioning could undergo an equilibrium process in several seconds or minutes. I did not find description the duration time of signal collection process (but I guess it may in a very short time). So how do the authors consider the equilibrium process in bringing uncertainty to the current experimental results?

**(3)** The authors only selected 1:1 molar ratio for carboxylic acids (C1-C6, vertical) and alkyl amines (A1-A6, horizontal) mixed solutions in this manuscript. Can the authors explain why chose 1:1 molar ratio for experiments as a represent? since I personally think only one ratio of the mixtures may not be sufficient. And I am not sure that when the molar ratio alters (e.g., 1:3 (i.e., carboxylic acids dominated) and 3:1 (i.e., alkyl amines dominated), the factor results may change (Line 51-52: …a factor of several hundred).

**Minor comments:**

**(4) Abstract:** I personally think that the abstract is too long; please condense it to clearly state the main conclusions and their atmospheric significance.

**(5) Tables:** The format of most tables (1, 3, 4, 5, 6, 7) should be adjusted since they are not fit the specific guidelines.

**(6) Line 51-52:** Please add specific molar ratio, since the authors only have 1:1 molar ratio mixtures' result and it is unclear for other ratios.

**(7) Line 64:** "(Mccormick and Ludwig, 1967)" is not appropriated based on your reference part. It should be "McCormick".

**(8) Figure 1 and Figure 4**: If there are multiple subfigures, they need to be numbered (e.g., A, B, …). Do not used top, middle, and bottom.

**(9) Figure 1:** How to explain that all peak positions for single organic (p1 and p2 in the first two subfigures) slightly moved to the right in the third subfigure (spectrum signals for mixed solutions). In addition, why peak 2 in subfigure (A) and subfigure (B) were different as they both represent the C atom in $CH_3$ group.

**(10) Table 1:** The number of decimal places retained should be consistent (at least for same row). In addition, the values in the 4th row (carbon-normalized $I_{norm}= I_{norm}/(x,y)$) seems to be wrong, for example, for C4 column, 11.5/3=3.83, your value is 2.87), please check the whole calculation and carefully revised the corresponding Figures and Tables.

**(11) Line 430-431:** The author should give some explanations on what caused the different molecular orientations for alkyl ammonium and carboxylate anions.

**(12) Line 516-517:** Please add references.

**(13) References section:** The format of the references is not consistent (e.g., some journal names are full but others are abbreviations). Please carefully check the format throughout the whole references based on the ACP guideline or other published paper.

Noziere, B., Baduel, C., & Jaffrezo, J. L. (2014). The dynamic surface tension of atmospheric aerosol surfactants reveals new aspects of cloud activation. *Nature Communications, 5*(1), 3335. https://doi.org/10.1038/ncomms4335

---

## Author Response (AR1)

**Referee A**

**General Assessment:**

**Reviewer:** The paper investigates the surface propensity of organic surfactant molecular ions in aqueous solutions, which is crucial for understanding atmospheric aerosol behaviour. It focuses on carboxylic acids and alkyl amines, common organics surfactants in aerosols, and measures their surface enrichment using liquid-jet photoelectron spectroscopy. The study reveals an exponential increase in surface propensity with chain length and a significant boost in surface enrichment due to cooperative ion-ion interactions in mixed solutions. Results show surface compositions can vastly differ from bulk, impacting atmospheric processes like droplet formation and chemical aging. This research enhances the understanding of aerosol surface phenomena needed for climate model. The topic fits scope of ACP. The paper can be considered for publication after revisions that address the following concerns.

**Our reply:** We thank the reviewer for the thoughtful reading of the manuscript and for this positive assessment.

**Major comments:**

**Reviewer:** (1) As the authors described that the experiments were all performed under 10 °C (solution), $2 \times 10^{-4}$ mbar (environment), So how do the authors think that the experiments could provide valuable information for atmospheric implication. I suggest the author give more explanations on temperature setting and improved the atmospheric implication part.

**Our reply:** The experiments were performed with the temperature of the solution bath set at 10 °C. The microjet assembly through which the solution was injected, was at (or slightly below) room temperature, which can thus be considered the temperature at the point of ejection. Evaporative cooling in vacuum decreases the temperature somewhat, and the sample temperature is estimated to be on the order of ~15-10 °C at the measurement (light-interaction) point of the liquid jet. While not fully comparable, we recently undertook systematic measurements of the cooling rate and corresponding jet temperature of a liquid flat jet based on infrared (temperature) imaging, which support these estimates; see: Buttersack et al., Struct. Dyn. 10, 034901 (2023).

Regarding the atmospheric relevance, temperatures in the range of ~10 °C are found in the atmosphere, both at ground level and higher up. The environmental pressures at these altitudes are certainly higher than what was used in the experiment. Yet, we argue that the evaporative conditions found in the experiment are not so different from conditions in the atmosphere where partial pressures are typically well below saturation. It is also important to note that the immediate liquid-jet surface region is essential close to the vapor pressure of the solvent, which then rapidly falls off with distance from the jet. We note that this pertains to the water solvent, the molecular ions do not have an appreciable vapor pressure.

To clarify these issues, and make the readers aware of them, we have added the following text to the beginning of the "Atmospheric implications" section (page 21 lines 608 ff.): *"Before elaborating on the atmospheric implications, we briefly discuss the conditions of our measurements and their atmospheric relevance. Our PES studies from aqueous solutions were performed near ~10 °C temperature and at rough vacuum conditions. Such temperatures are found in the atmosphere, both at ground level and higher up. Operation at low pressure implies that the measurements were made under evaporative conditions with the immediate surface being at or slightly below the vapor pressure of water, which are also found in the atmosphere. We note that this pertains to the water solvent, while the investigated molecular ions do not have an appreciable vapor pressure. Based on these considerations, we are confident that the phenomena we report are relevant for atmospheric aerosols, and we will now proceed to elaborate on the atmospheric implications."*

**Reviewer:** (2) As reported in the literature (for example, Figure 1 in Noziere, Baduel, and Jaffrezo (2014), surfactant delayed surface tension equilibrium, indicating that bulk-to-surface partitioning could undergo an equilibrium process in several seconds or minutes. I did not find description the duration

time of signal collection process (but I guess it may in a very short time). So how do the authors consider the equilibrium process in bringing uncertainty to the current experimental results?

**Our reply:** The time from the liquid jet exiting the capillary until it reaches the interaction point (of ionization) can be estimated from the jet velocity, ~30 m/s, and the distance, ~$10^{-3}$ m, to be in the order of ~$10^{-5}$ s. This is indeed much shorter than the timescales reported in Noziere, Baduel, and Jaffrezo (2014). Those latter results were obtained for significantly larger solutes; for the smaller molecules in the present study shorter equilibration times are to be expected. One indirect support for this comes from molecular dynamics simulations of the C6/A6 system, for which an equilibrium situation at the surface is reached with simulation times in the range of tens of nanoseconds, see e.g., Ref. (Ekholm et al., 2018). Insufficient equilibration time is thus likely less of an issue for surfactants with shorter chain length.

The timescale of the extended equilibration processes, reported by Noziere, Baduel, and Jaffrezo (2014), as well as other authors, is not limited by the diffusion. The determining factor for the timescale may instead be a barrier for surface adsorption caused by repulsion of molecular ions in the solution by similarly charged ions already at the surface. This slowing down of the equilibration processes seems to be significantly reduced by the presence of small inorganic ions, which efficiently reduce the repulsion between the molecular ions by screening. It may also be possible that such a barrier for surface adsorption caused by repulsion between molecular ions has less of an effect on the timescale of the equilibration processes in the present cases with molecular ions of both positive and negative charge. To investigate this is beyond the scope of the paper, and will have to be subject to future research.

To summarize, if full equilibration has not been reached in our measurement, the ultimate surface coverage would be higher than what we determine. We conclude that the observed cooperative enhancement of the surface coverage is not affected by slow equilibration times. On the contrary, it is most likely larger than what we detect.

To clarify this, we have added the following text on page 12, lines 382 ff.: *"It should be noted that the measurement is performed at approximately ~$10^{-5}$ s after the liquid jet has exited the capillary (jet velocity ~30 m/s, distance ~$10^{-3}$ m). While this is a long time compared to the tens of nanoseconds used in MD simulations to obtain equilibrium conditions, see, e.g., Ref. (Ekholm et al., 2018) for the C6/A6 system, molecules with yet longer chain lengths may lead to further surface enrichment on a longer time scale. We cannot draw a sharp line and therefore conclude that the results presented here might be considered the lower limit for the observed surface behavior."*

**Reviewer:** (3) The authors only selected 1:1 molar ratio for carboxylic acids (C1-C6, vertical) and alkyl amines (A1-A6, horizontal) mixed solutions in this manuscript. Can the authors explain why chose 1:1 molar ratio for experiments as a represent? since I personally think only one ratio of the mixtures may not be sufficient. And I am not sure that when the molar ratio alters (e.g., 1:3 (i.e., carboxylic acids dominated) and 3:1 (i.e., alkyl amines dominated), the factor results may change (Line 51-52: …a factor of several hundred).

**Our reply:** The 1:1 molar ratio was chosen so as to have the same ratio for all mixed solutions, thereby facilitating their comparison. This has allowed us to demonstrate the cooperative enhancement of the coverage and its dependence on variable alkyl chain length. We agree with the referee that other molar ratios may alter the values of the enhancement factors. In particular, we would expect a somewhat smaller surface enrichment for asymmetric ratios, since then the ion pairing effect leading to the surface enhancement is reduced. Thus, the results of our study could be seen as the extreme case of this effect. To explore different molar ratios was beyond the scope of the presently reported investigation and could be subject for future research. We have made no associated change in the text.

**Minor comments:**

**Reviewer:** (4) **Abstract:** I personally think that the abstract is too long; please condense it to clearly state the main conclusions and their atmospheric significance.

**Our reply:** We have cut the last part of the abstract where we had further detailed the findings of the present work, and instead expand on the atmospheric significance of or results. The new text (pages 2, lines 52-56) now reads: *"These changes can significantly influence radiative forcing via aerosol growth, cloud condensation nuclei activity, and aerosol chemical aging. Our results demonstrate the principal feasibility of a more advanced input of molecular details for creating parameterized descriptions of aerosol surface composition needed to properly account for their impacts in climate models."*

*We deleted this part: "~~with pronounced chain length-dependent variations. Our results demonstrate that the surface composition of these water-organics systems can be very different from the bulk composition, and that the surface compositions of organic mixtures cannot be directly inferred from the behaviour of the single components. For aerosols containing these or similar species, this means that all surface-related properties and processes will be enhanced, and implications for atmospherically relevant processes such as water accommodation, droplet formation, and chemical aging, are discussed~~."*

**Reviewer:** (5) **Tables:** The format of most tables (1, 3, 4, 5, 6, 7) should be adjusted since they are not fit the specific guidelines.

**Our reply:** We assume the reviewer refers to the following guideline; *"Horizontal lines should normally only appear above and below the table, and as a separator between the head and the main body of the table."* We have modified the tables accordingly.

**Reviewer:** (6) **Line 51-52:** Please add specific molar ratio, since the authors only have 1:1 molar ratio mixtures' result and it is unclear for other ratios.

**Our reply:** This change has been made. The sentence on page 2 (line 49) now reads: *"An exponential increase in surface propensity is found for the single-species solutions, with cooperative solute-solute effects in mixed solutions of 1:1 molar ratio drastically increasing the number of molecules present at the solutions' surfaces up to a factor of several hundred."*

**Reviewer:** (7) **Line 64:** "(Mccormick and Ludwig, 1967)" is not appropriated based on your reference part. It should be "McCormick".

**Our reply:** (Mccormick and Ludwig, 1967) has been changed to (McCormick and Ludwig, 1967).

**Reviewer:** (8) **Figure 1 and Figure 4**: If there are multiple subfigures, they need to be numbered (e.g., A, B, …). Do not used top, middle, and bottom.

**Our reply:** While we consider this a matter of personal preference, we have made requested changes in the figure and in the text accordingly.

**Reviewer:** (9) **Figure 1:** How to explain that all peak positions for single organic (p1 and p2 in the first two subfigures) slightly moved to the right in the third subfigure (spectrum signals for mixed solutions). In addition, why peak 2 in subfigure (A) and subfigure (B) were different as they both represent the C atom in $CH_3$ group.

**Our reply:** The kinetic energies in the PES measurements are shown "as measured" without any calibration or post-alignment done. Mentioned small shifts can be due to, e.g., changes in work function of the solution as the surface composition is changed. The magnitude of these shifts is small, and does not affect the identification of the spectral peaks.

We have added on page 9 (lines 300 ff.) in the main text the following sentence detailing the origin of the two energy axes shown in the figure: *"The bottom axis presents the as-measured kinetic energies (KE) of the photoelectrons, and the respective (uncalibrated) electron binding energies (BE), calculated as photon energy minus KE, are given on the top energy axis."*

In addition to the brief explanation of the observed various chemical shifts in the first paragraph of Results and Discussion, we have extended the caption of Figure 1 to further explain the origin of relative shifts (in particular, in response to point 5 of the other Reviewer). It now reads on pages 11, lines 352 ff.: *"Relative to the alkyl-like carbons (CH$_2$ and CH$_3$, peaks p2), the signals from the carbons in the two functional groups, -CH$_2$-NH$_3^+$ and -COO$^-$, peaks p1, are found at higher binding energies. On a qualitative level, this shift is caused by the proximity of electronegative atoms, which reduce the electron density around the carbon atoms in the initial state, and thereby also the final state screening. This leads to a higher binding energy as compared to carbons bound to hydrogens or other carbons. In addition, the net charge associated with the two functional groups will lead to a shift. Moreover, small energy shifts associated with changes in work function of the solution as the surface composition is changed may occur which, however, does not affect the identification of the spectral peaks. For a further discussion, see e.g., Ref. (Werner et al., 2018), with which the present results are consistent, as well as Ref. (Thürmer et al., Chem. Sci. 2021)."*

**Reviewer:** (10) **Table 1:** The number of decimal places retained should be consistent (at least for same row). In addition, the values in the 4$^{th}$ row (carbon-normalized I$_{norm}$= I$_{norm}$ /(x,y)) seems to be wrong, for example, for C4 column, 11.5/3=3.83, your value is 2.87), please check the whole calculation and carefully revised the corresponding Figures and Tables.

**Our reply:** First, we would like to note that the value of 2.87 is correct, since I$_{norm}$= I$_{norm}$ /(x,y) implies 11.5/4 for C4. The reviewer might have confused the carbon number x with the effective carbon number k = x-1. However, we have found a mistake in the column for C6, where the intensity values for peaks p1 and p2 were wrong. We have corrected the values in the Table accordingly.

**Reviewer:** (11) **Line 430-431:** The author should give some explanations on what caused the different molecular orientations for alkyl ammonium and carboxylate anions.

**Our reply:** The molecular orientation at the surface is due to a balance between the hydrophobic and the hydrophilic interactions. This can be investigated by, e.g., molecular-dynamics simulations, which may be subject for further research. Some work on individual groups exists, such as simulations for the carboxylate anions by Houriez et al., (J. Phys. Chem. B 2015, 119, 36, 12094–12107) exist, which confirms a strong orientation perpendicular to the water surface. Yet, to our knowledge, there is no comparative theoretical study for alkyl ammonium and carboxylate ions. To provide a qualitative discussion of the reasons for the different molecular orientations of these ions, we have added the following text (page 14, lines 437 ff.): *"The molecular orientation at the surface is due to a balance between the hydrophobic and the hydrophilic interactions. The charged groups interact strongly with the water and tend to be submerged, i.e., fully surrounded by the solvent; here, carboxylate interacts stronger with water than ammonium. In contrast, the alkyl chains can only interact weakly with the water and tend to be expelled from the liquid interface. We speculate that for the alkyl carboxylates this could result in a somewhat more deeply solvated carboxylate group with the alkyl chain pointing outwards, while for the alkyl ammonium the less deeply solvated ammonium group would allow the alkyl chain to orient itself along the surface for increased bonding to the outermost water molecules. One may also expect that the counter ions, Na$^+$ or Br$^-$, may have a small contribution to the emerging surface structure but that aspect has not been explored in the present study."*

**Reviewer:** (12) **Line 516-517:** Please add references.

**Our reply:** The references (Walz et al., 2015; Walz et al., 2016) have been added.

**Reviewer:** (13) **References section:** The format of the references is not consistent (e.g., some journal names are full but others are abbreviations)**.** Please carefully check the format throughout the whole references based on the ACP guideline or other published paper.

**Our reply:** We have thoroughly revised the references.

**Referee B**

**General Assessment:**

**Reviewer:** In the paper *Strongly Enhanced Cooperative Surface Propensity of Atmospherically Relevant Organic Molecular Ions in Aqueous Solution* by Kaur et al., the authors use liquid jet photoelectron spectroscopy that is tuned specifically to the interfacial region of the liquid to study the cooperative effects of different organic acids and bases dissolved in solution. The authors use the salt versions of both the acid and base species with counter ions of sodium and bromide, respectively, and create neat and mixed solutions of each species to study whether there is enhancement of either species near the interface. They use the relative ratios between the distinct carbon species (C1 is the carboxyl group on the acid while C1 is attached to the amine on the base) to argue that there is enhancement of the groups at the surface – of which the impact and implications for atmospheric reactions that rely on specific ions or molecules near the interface is significant due to the overwhelming amount of carbon that now resides near the surface blocking the ions that are needed to carry out these reactions further into the bulk of the solution.

**Our reply:** We appreciate this short summary.

**Comments and Questions**

There are several questions and comments I would like the authors to address:

**Reviewer:** (1) The authors rely heavily on ratios of ratios to make their arguments throughout the manuscript. There is limited discussion on the error that would be propagated throughout these measurements until Table 6. Can the authors comment on the relative error of the ratios they are taking? The fits do look very good but there seems to be, by eye, a larger error in the fit especially in the A6 spectra, which seem to have the highest enhancements.

**Our reply:** All peak intensities are normalized to the value of formate (C1); see Methods. The C1 signal intensity can be determined very accurately and was confirmed in repeat measurements. Normalization by carbon number does not introduce additional uncertainties. Furthermore, if we take the intensity ratio of two components, it does not matter whether the intensities have been normalized beforehand, since the normalization term cancels out. This implies that the uncertainty of the total signal intensities largely arises from the accuracy with which individual peak components can be determined from the mixed solutions. This is of no issue for Figures 2 and 3 as well as Tables 2, 3 and 7, where we simply use the total intensity of all components. Table 4 and 6 use combined intensities (the sum of peaks p1 and p2) for each component to form a ratio, and we explicitly stated an error for this calculation. The only result, where larger errors may play a role are the ratios in Table 5. The strongly overlapping signal of peak p2 was a challenge for the mixtures with A6. For this reason, we refrain from giving a fit result for the Cx/A6 mixtures and only state an average. After careful error analysis, we now state an error of $\pm 0.1$ for all other values in this table. We also opted to state the error estimate more clearly in each table caption.

**Reviewer:** (2) Given the authors use multiple ratios, why have they not taken a ratio of the C1s to O1s at similar probe depths? This would be a direct measure of the concentration which could be compared to the solution bulk concentration. Are there possible pitfalls in this ratio where the authors are more confident in their method over this for investigating the enhancement?

**Our reply:** We have two reasons for using the C 1s intensity rather than the C 1s / O 1s ratio to determine the surface enhancement. First, the O 1s signal is dominated by contributions from the solvent (water). It may be possible to separate the contribution of the solute to some extent by fitting the spectra, but not without introducing difficult-to-quantify uncertainties. Second, the existence of an organic overlayer will affect the signal from the bulk water by scattering of the outgoing electrons. This was demonstrated by the changing signal of the water valence band in the supporting information. It would be a difficult

undertaking to model these changes, and certainly not without introducing considerable uncertainties. We have addressed this issue in the revision on page 5, line 143-144: *"O 1s signal intensities were not considered due to the strong overlap of this spectral region with the solvent."*

**Reviewer:** (3) The authors argue that a deviation larger than the ideal is related to the orientation of the molecular species being perpendicular to the interface while a lower deviation is due to a parallel orientation. Have the authors considered that this could also be due to the negative and positive charges that are present on the functional groups rather than an orientation effect? The solvation of the negative charge of the carboxyl group would be different in aqueous solution compared with the positive charge of the amine group. The hydrogen bonding at these functional groups may be the cause of the deviation from ideal rather than the orientation.

**Our reply:** The negative and positive charges of the functional groups will affect the binding energies, not the intensities. On the other hand, the charges affect the molecular orientations at the surface, which are due to a balance between the hydrophobic and the hydrophilic interactions; also see our reply to the other reviewer's point 11. To provide a qualitative discussion of the reasons for the different molecular orientations of alkyl ammonium and carboxylate anions, we have added the following text (page 14, lines 437 ff.): *"The molecular orientation at the surface is due to a balance between the hydrophobic and the hydrophilic interactions. The charged groups interact strongly with the water and tend to be submerged, i.e., fully surrounded by the solvent; here, carboxylate interacts stronger with water than ammonium. In contrast, the alkyl chains can only interact weakly with the water and tend to be expelled from the liquid interface. We speculate that for the alkyl carboxylates this could result in a somewhat more deeply solvated carboxylate group with the alkyl chain pointing outwards, while for the alkyl ammonium the less deeply solvated ammonium group would allow the alkyl chain to orient itself along the surface for increased bonding to the outermost water molecules. One may also expect that the counter ions, $Na^+$ or $Br^-$, may have a small contribution to the emerging surface structure but that aspect has not been explored in the present study."*

**Reviewer:** (4) The authors say that the small counter ions would be impacted by the organic groups at the surface. Did they look into this during their experiments? The counter ions of both $Na^+$ and $Br^-$ were used with Br being very atmospherically relevant.

**Our reply:** The surface propensity of the small counter ions was not measured in the experiments. Our statement that they would be impacted by the organic groups at the surface is based on published reports of such effects for similar systems, see, e.g., Ref. (Gopakumar et al., 2022). Further investigation along these lines could be subject for future studies. In any case, the text added for point 3 above closes with the following sentence to address this comment (see page 15, top): *"One may also expect that the counter ions, $Na^+$ or $Br^-$, may have a small contribution to the emerging surface structure but that aspect has not been explored in the present study."*

We furthermore now provide a reference to support the following sentence / expectation on page 21 (line 605): *"As a result, the inorganic counter ions can be expected to exhibit reduced enrichment in the sub-layer of such mixed cationic–anionic molecular ion solutions (compare Figure 4 and Ref. (Gopakumar et al., 2022)), and hence fewer ions are available for reactions with gas-phase species."*

**Reviewer:** (5) The authors did not discuss much about the binding energy shifts in the C1s region. Although the absolute energies are difficult to pin down, the relative binding energy shifts between the carbons might be useful. Did the authors look into this at all? A concern of mine is that there may be pH shifts between the carboxyl and amine groups that could show up in the relative binding energy shifts even though the salts version of the organics is used. Can the authors comment on this? Did the authors take a bulk pH measurement of each solution prior to the experiment?

**Our reply:** We chose to not elaborate on the C1s binding energy shifts for two reasons. First, our results are consistent with the published results by Werner et al., 2018. Second, the focus of the paper is on the surface propensity, for which the measured C1s intensities rather than the binding energies are important. Energy-calibrating the spectra would add considerable measurement time to the already quite laborious experiment. In the revision we have added a brief explanation on the origin of the shifts (without losing focus) to the caption of Figure 1. It now reads on pages 11, lines 352 ff.: *"Relative to the alkyl-like carbons (CH$_2$ and CH$_3$, peaks p2), the signals from the carbons in the two functional groups, -CH$_2$-NH$_3^+$ and -COO$^-$, peaks p1, are found at higher binding energies. On a qualitative level, this shift is caused by the proximity of electronegative atoms, which reduce the electron density around the carbon atoms in the initial state, and thereby also the final-state screening. This leads to a higher binding energy as compared to carbons bound to hydrogens or other carbons. In addition, the net charge associated with the two functional groups will lead to a shift. Moreover, small energy shifts associated with changes in work function of the solution as the surface composition is changed may occur, which, however, does not affect the identification of the spectral peaks. For a further discussion, see e.g., Ref. (Werner et al., 2018), with which the present results are consistent, as well as Ref. (Thürmer et al., Chem. Sci. 2021)."*

Regarding the solutions' pH: The binding energy shifts are known to be quite independent of pH for each protonation state of each species. Large shifts are observed upon protonation / deprotonation, *i.e.*, around the p$K_a$. The solution pH was in all cases far from the respective p$K_a$ values. Moreover, any such protonation/deprotonation would result in neutral species (carboxylic acid and alkyl amine), which have significantly shifted binding energies and higher surface propensities, see, *e.g.*, Ref. Werner et al., 2018. The PE spectra are thus very sensitive to neutral species, but no indications of such were observed.

**Reviewer:** (6) Do the authors believe that the enhancement of the mixed long chain organics (C6/A6) is due to the change in orientation of the A6 molecule near the interface?

**Our reply:** We consider the two effects to be connected. By changing the orientation from lying down to standing up, a higher molecular density is enabled. This is discussed in the "Surface Orientation" section, and now clarified with the following added sentence on page 18 (line 533): *"The molecular-scale mechanism of the cooperative surface propensity may thus include changes of the orientation from lying down to standing up, enabling higher molecular surface densities."*

**Reviewer:** (7) Have the authors done a concentration dependence measurement to look for saturation of any of the organics they studied?

**Our reply:** All the measurements were done for a total concentration of 0.1 M of organics. This means 0.1 M of Cx or Ax for the unmixed samples, and 0.05 M of Cx + 0.05 M of Ax for the mixed cases. These concentrations were chosen such that the unmixed samples would be far from surface saturation. A6 and C6 have the highest surface propensities among the single species, and at 0.1 M, the surface coverage of A6 is ~0.37 and of C6 ~0.15 of the maximum coverage, *i.e.*, well below surface saturation (Ekholm, 2018). The magnitude of cooperative enhancement of the coverage may mean that the most enriched combination, A6/C6, is somewhat affected by surface saturation. An extended investigation of the concentration dependence would certainly be interesting to do in the future, but is beyond the scope of the paper.

**Reviewer:** (8) I found some of the language used in the manuscript to create a lot of skepticism in the assertions. The authors use a lot of approximates and relative terms rather than stating what they have and discussing the errors. Perhaps with more thought in their error analysis they will feel more comfortable using stronger language in their observations and fits. This a minor comment to strengthen the manuscript overall.

**Our reply:** Correlating electronic structure with geometric structure is in many cases not straight-forward, typically requiring supportive theoretical simulations. This is certainly the case for the large

molecules studied here and their interaction in a complex aqueous environment, leading to a large increase of surface propensity, Hence, a safe interpretation of the data presented will inevitably base on model assumptions. Yet, with the inclusion of an extended error analysis as mentioned above, we have taken this opportunity to strengthen several statements in the revised manuscript.